# Pleiotropic Effects of Exosomes as a Therapy for Stroke Recovery

**DOI:** 10.3390/ijms21186894

**Published:** 2020-09-20

**Authors:** Yuji Ueno, Kenichiro Hira, Nobukazu Miyamoto, Chikage Kijima, Toshiki Inaba, Nobutaka Hattori

**Affiliations:** Department of Neurology, Juntendo University Faculty of Medicine, Tokyo 113-8421, Japan; ra-hi-.com@live.jp (K.H.); nobu-m@juntendo.ac.jp (N.M.); c-kijima@juntendo.ac.jp (C.K.); to-inaba@juntendo.ac.jp (T.I.); nhattori@juntendo.ac.jp (N.H.)

**Keywords:** ischemic stroke, exosomes, recovery, axonal outgrowth, neurogenesis, mesenchymal stromal cells

## Abstract

Stroke is the leading cause of disability, and stroke survivors suffer from long-term sequelae even after receiving recombinant tissue plasminogen activator therapy and endovascular intracranial thrombectomy. Increasing evidence suggests that exosomes, nano-sized extracellular membrane vesicles, enhance neurogenesis, angiogenesis, and axonal outgrowth, all the while suppressing inflammatory reactions, thereby enhancing functional recovery after stroke. A systematic literature review to study the association of stroke recovery with exosome therapy was carried out, analyzing species, stroke model, source of exosomes, behavioral analyses, and outcome data, as well as molecular mechanisms. Thirteen studies were included in the present systematic review. In the majority of studies, exosomes derived from mesenchymal stromal cells or stem cells were administered intravenously within 24 h after transient middle cerebral artery occlusion, showing a significant improvement of neurological severity and motor functions. Specific microRNAs and molecules were identified by mechanistic investigations, and their amplification was shown to further enhance therapeutic effects, including neurogenesis, angiogenesis, axonal outgrowth, and synaptogenesis. Overall, this review addresses the current advances in exosome therapy for stroke recovery in preclinical studies, which can hopefully be preparatory steps for the future development of clinical trials involving stroke survivors to improve functional outcomes.

## 1. Introduction

Stroke is the second leading cause of death and the leading cause of acquired disability worldwide, and more than 40 million patients suffered from stroke according to the Global Burden of Diseases, Injuries, and Risk Factors 2015 study [1]. Furthermore, its incidence is increasing because of population aging, and younger people are also affected in developing countries [2]. With approximately 800,000 new stroke victims in the United States each year, the limited degree of spontaneous recovery after a stroke causes a large personal and societal burden in terms of lost productivity, lost independence, and social withdrawal [3]. Thrombolysis with recombinant tissue plasminogen activator (rt-PA) and endovascular intracranial thrombectomy (EVT) are currently the two major treatments for the acute stage of ischemic stroke, resulting in reduction of mortality and amelioration of functional outcomes three months after stroke [4,5]. However, these approaches have been limited by an insufficient therapeutic window (rt-PA, <4.5 h; EVT <6 h), and ≈40% of stroke survivors undergoing these therapies still have poor functional outcomes [6]. Furthermore, rt-PA appears to be associated with relative increases of intracranial hemorrhagic complications [7,8]. Edaravone, a potent free radical scavenger, approved in Japan in 2001 for the treatment of acute ischemic stroke, has also been used in China and other Asian countries, and it has been shown to improve neurological deficits, but it has not been widely used worldwide [9]. Other than pharmacological therapy, in-hospital stroke unit care with multidisciplinary teams has been shown to reduce mortality, dependency, and poor functional outcomes [10]. Despite these attempts, stroke remains a significant source of long-term neurological disability, and therapies to enhance and improve functional recovery are urgently needed.

Restorative and regenerative therapies to amplify endogenous processes for tissue repair including neurogenesis, angiogenesis, and axonal outgrowth after stroke would be promising as a novel therapeutic strategy. Treatment agents must be delivered to the site of ischemic injury, and many attempts have been made by intravenous injection and stereotactic and intraventricular administration [11,12,13]. In patients with stroke, axons decrease in the peri-infarct area, but they start to regrow at the end of the acute stage, and they are substantially increased in the chronic phase to as much as intact levels [14]. Therefore, it is essential to administer these agents in the acute phase to reverse molecular cascades leading to necrosis and apoptosis, whereas therapies in the subacute and chronic phases enhance molecular cascades related to axonal outgrowth. There have been many studies of neurorestorative and neuroregenerative therapies for the treatment of ischemic injury using cell-based therapy. Cell-based therapy facilitates not only neurogenesis and angiogenesis [15,16,17], but also axonal regeneration, together with oligodendrogenesis [18,19]. On the other hand, pharmacological therapies such as the inhibition of NogoA also enhance axonal outgrowth [11]. Previous studies examined whether Phosphatase and Tensin Homolog Deleted from Chromosome (PTEN)/protein kinase B (Akt)/Glycogen Synthase Kinase 3 beta (GSK-3β) signaling were implicated in the mechanisms of axonal outgrowth in rodent ischemic brains [14,20]. Furthermore, we demonstrated that inhibition of semaphoring 3A from the acute to the subacute phase of stroke in the peri-infarct area increased axonal outgrowth related to the Rho family GTPase 1 (Rnd1)/R-Ras/Akt/GSK-3β pathway, together with suppression of glial fibrillary acidic protein (GFAP)-immunoreactive astrocytes, and functional recovery was promoted in the chronic stage of stroke [13]. Additionally, it was shown that PTEN/Akt/mammalian target of rapamycin (mTOR) signaling was implicated in the mechanisms of axonal outgrowth after L-carnitine therapy in a rat model of chronic cerebral hypoperfusion [21].

Exosomes, extracellular vesicles that are 40–100 nm in diameter, contain microRNAs, mRNAs, lipids, and proteins, and they are released via the endocytic pathway from many kinds of cells [22,23]. There has been considerable interest in exosomes contributing to the transfer of these molecules and genetic information to target cells, causing alterations of cellular activities and functions under physiological and pathophysiological conditions [23,24]. Treatment with exosomes has been proven to be a good candidate for acute kidney injury, as well as myocardial injury, by inhibiting the inflammatory reaction and oxidative stress, regulating autophagy, and enhancing angiogenesis and tissue repair [25,26,27,28,29,30]. In the central nervous system, exosomes exert important roles in cell-cell communication in brain remodeling after stroke [31]. The therapeutic effects of exosomes have been attributed to their surface markers, molecular contents, ability to cross the blood–brain barrier (BBB), and roles in mediating neural regeneration. Emerging data indicate that exosomes derived from mesenchymal stromal cells (MSCs) and stem cells enhance stroke recovery [32]. In the current systematic review, recent insights into the role of exosomes and their effects of altering molecular cascades in neuroregeneration and brain repair processes after stroke are highlighted, and potential applications of exosomes for stroke therapy are discussed. So far, exosome therapy has not been applied in a clinical setting, and only a few clinical trials have been conducted in stroke patients (https://clinicaltrials.gov). Thus, the therapeutic efficacy and safety of exosomes are essentially unknown in humans. Therefore, a systematic review to explore the treatment effects of exosomes for functional recovery after stroke in preclinical studies using animal stroke models was performed.

## 2. Methods

To explore the association of exosome therapy with functional recovery after stroke, a systematic review that followed the Preferred Reporting Items for Systematic Reviews and Meta-analyses (PRISMA) guidelines was conducted [33]. The search was restricted to articles written in English, and it was performed using PubMed and Medline by entering the search terms “stroke” or “ischemia”, “recovery”, and “exosomes”. Review articles, meta-analyses, case reports, editorial comments, letters, meeting abstracts, and studies not fulfilling the inclusion criteria for their content were excluded. The literature search was continued until no further publications were identified. The main outcome measures that have been considered were mainly related to motor function and neurological deficits. In part, additional outcomes, such as infarct area, molecular pathways including microRNA, mRNA, and protein, and inflammatory markers or systemic cytokine levels, have also been assessed. Because the current systematic literature search focused on functional stroke recovery after exosome therapy, in vitro and in vivo studies without a physiological analysis were excluded. In vivo studies with physiological analysis that evaluated the acute stage (<7 days after stroke) and studies using exosomes not derived from cells (e.g., MSCs, stem cells) were also excluded. The stroke model was restricted to transient or permanent focal cerebral ischemia, such as occlusion of the middle cerebral artery, but not ligation or stenosis of bilateral common carotid arteries and a perinatal model of cerebral hypoxia-ischemia [21,34]. Two investigators (K.H. and Y.U) independently screened each title and abstract. Studies published by 15 April 2020 were considered. In the case of disagreement regarding study eligibility, a consensus meeting was arranged to resolve potential discrepancies by open discussion.

## 3. Results

The database literature search identified 357 papers. Four additional publications were found after screening the reference lists. Of these 361 publications, 299 papers were excluded after abstract review for inappropriate content. The 62 remaining articles were reviewed on a full-text basis. Furthermore, 49 of them were excluded due to different study designs (*n* = 19), other violation of the inclusion criteria (*n* = 8), and different outcomes (*n* = 22). Finally, 13 studies met the eligibility criteria and were included in the review. Details of the search and article selection are summarized in the flow diagram (Figure 1).

## 4. Subjects in Included Studies

All 13 included studies were animal studies, involving mice in one and rats in 12 (Table 1). In them, a transient middle cerebral occlusion (tMCAO) model for 30, 50, 60, 90, and 120 min was used in 1, 1, 1, 2, and 8 studies, respectively. The sources of exosomes were bone marrow stromal cells (BMSCs), human umbilical cord blood MSCs (HUCB-MSCs), adipose-derived mesenchymal stem cells (ADMSCs), urine-derived stem cells, and rat adipose-derived stem cells in 8, 2, 1, 1, and 1 studies, respectively. Intravenous administration was performed in 12 studies, whereas the one remaining study used stereotaxic administration. Timing of exosome administration varied from immediately to 24 h after reperfusion, and dosage ranged from 30 to 150 μg. Behavioral assessments including the modified neurological severity score (mNSS), the foot-fault test, modified adhesive removal test, beam walking, shuttle-box test, accelerating rotarod performance tests, elevated body swing test, Garcia score, tightrope test, and the cylinder and ladder rung walking test were performed, and the maximum time of evaluation of motor function was 7–28 days.

## 5. Therapeutic Effect of Exosomes for Stroke Recovery

### 5.1. Exosomes Derived from MSCs

Xin and colleagues first extracted exosomes derived from multipotent MSCs in bone marrow from adult male Wistar rats. Using a rat model of tMCAO for 2 h, exosomes derived from BMSCs were injected 24 h after stroke via the tail vein, and neurophysiological analyses such as the foot-fault test and mNSS were conducted at 1, 3, 7, 14, 21, and 28 days after tMCAO. At 14, 21, and 28 days, BMSC-generated exosome-treated rats were significantly improved with respect to neurological deficits on the mNSS and motor function on the foot-fault test (*p* < 0.05). At 28 days after MCAO, there were no significant differences in ischemic lesion volume between the exosome-treated rats (31.1% ± 3.79%) and the vehicle-treated rats (32.9% ± 3.34%). On immunohistochemistry of the peri-infarct area, axonal outgrowth and myelination and synaptogenesis were increased after the exosome treatments. Moreover, tMCAO rats treated with exosomes showed increases in the percentages of doublecortin (DCN)- and 5-bromodeoxyuridine (BrdU)-positive cells and of von Willebrand- and BrdU-positive cells. These data indicated that BMSC-derived exosomes improved motor recovery in ischemic rats, along with histological alterations with neurogenesis, angiogenesis, and synaptogenesis [35].

Zhao and colleagues studied the anti-inflammatory effects of BMSC-generated exosomes for acute cerebral ischemia. They treated rats subjected to tMCAO for 90 min with BMSC-derived exosomes at 2 h after ischemia. Treatment with BMSC-derived exosomes significantly improved neurological deficits and performance on the shuttle-box test as an evaluation of locomotor activity at 7 days after tMCAO [36]. BMSC-derived exosomes also significantly suppressed M1 microglial polarization and increased M2 microglial cells, together with reduction of pro-inflammatory cytokines, such as tumor necrosis factor (TNF)-α, interleukin (IL)-1β, and IL-12, and upregulation of anti-inflammatory molecules, such as brain-derived neurotrophic factor, transforming growth factor-β, and IL-1, and thereby downregulated extracellular signal regulated kinase 1/2 phosphorylation [36].

Doeppner and colleagues examined the therapeutic effects of exosomes derived from BMSCs compared with BMSCs in focal cerebral ischemia in mice. Two independent BMSC lines from human donors and their corresponding BMSC-generated exosome fractions were administered at 1, 3, and 5 days after focal cerebral ischemia, respectively, to mice. On the rotarod test, tightrope test, and corner turn test, exosomes derived from human-derived MSCs showed improvement of motor function at 7, 14, and 28 days after ischemia. Neuronal cell damage was alleviated, and neurogenesis and angiogenesis were enhanced by treatment with exosomes derived from human BMSCs. With regard to immunological processes after stroke, BMSC-generated exosomes did not modulate cellular infiltration in infarcted tissues, but they significantly altered the number of B lymphocytes, T lymphocytes, and natural killer cells, indicating that systemic treatment with BMSC-generated exosomes modulated the external environment appropriately to facilitate brain remodeling [37].

Ling et al. studied the effect of exosomes from human urine-derived stem cells (USCs) on neurogenesis after stroke. Treatment with USC-generated exosomes significantly improved mNSS scores and motor function on the foot-fault test from 14 to 28 days and infarct volume at 28 days in tMCAO for 2 h [38]. In particular, USC-generated exosomes enhanced proliferation and differentiation of neural stem cells (NSCs) in the peri-infarct subventricular zone in the tMCAO rat model. Furthermore, it was shown in vitro that proliferation and differentiation of NSCs due to USC-generated exosomes may be attributed to transfer of miR-26a for histone deacetylase 6 (HDAC6) inhibition [38].

In a study by Nalamolu and colleagues, using Sprague-Dawley rats with transient MCAO for 2 h, exosomes derived from HUCB-MSCs that were injected intravenously immediately after reperfusion reduced infarct volume. However, motor functions on the mNSS, modified adhesive removal test, beam walking, and accelerating rotarod performance test were exacerbated after stroke in exosome-treated rats [39]. Subsequently, they conducted another study using the same rat stroke model and showed that exosomes derived from the combination of normal and hypoxic MSCs significantly improved motor function on such tests [40]. It was suggested that differences in the type and expression levels of proteins and microRNAs in the exosomes from the combination of normal and hypoxic HUCB-MSCs could have contributed to the therapeutic benefit.

Furthermore, Moon reported that treatment with exosomes derived from BMSCs was superior to treatment with BMSCs themselves. As for mortality rates up to 14 days after stroke, the rate following treatment in the rat BMSC-derived extracellular vesicle group was notably lower (5%) than that of the other groups: human BMSCs (17%), PBS (20%), and fibroblast cell-derived extracellular vesicles (37%). On neurological function evaluation, the rat BMSC-derived extracellular vesicle treatment significantly reduced mNSS scores compared to treatments with PBS and fibroblast cell-derived extracellular vesicles at 14 days after tMCAO. In a cylinder test to evaluate forelimb deficits, there was a trend toward significance showing that 30 μg of rat BMSC-derived extracellular vesicles increased percent usage of the impaired forelimb compared to PBS treatment [41]. On immunohistochemistry, treatment with rat BMSC-derived extracellular vesicles significantly increased Ki67^+^DCN^+^ cells and von Willebrand cells, compared with PBS treatment and in a dose-dependent manner in the peri-infarct area 14 days after tMCAO, indicating that rat BMSC-derived extracellular vesicles facilitated neurogenesis and angiogenesis in the subacute phase of stroke. In vitro, miR-210, miR-184, and miR-137 were upregulated in rat BMSC-derived extracellular vesicles compared to fibroblast cell-derived extracellular vesicles. Furthermore, transfection with miR-210, as well as treatment with rat BMSC-derived extracellular vesicles, enhanced tube formation, together with reduction of ephrin-A3 in human umbilical vein endothelial cells. Transfection with miR-184 and treatment with rat BMSC-derived extracellular vesicles for human neural stem cells increased the proliferation of cells, together with downregulation of *Numbl*. These data indicated that miR-184 and miR-210, which are implicated in rat BMSC-derived extracellular vesicles, may be associated with therapeutic efficacy for neurogenesis and angiogenesis, respectively, after stroke. Finally, they concluded that BMSC-derived exosomes could be a candidate as alternate BMSCs for stroke treatment, promoting neurogenesis and angiogenesis, thereby improving motor function, and overcoming cell-associated limitations by stem cell therapy [41].

Safakheil et al. studied the effect of monotherapy with exosomes derived from BMSCs and combination therapy of BMSC-generated exosomes and rosuvastatin in Wistar rats subjected to transient MCAO. Monotherapy with exosomes, as well as combination therapy, decreased cell death together with downregulation of NOD-like receptor family pyrin domain containing 1 (NLRP1) and NOD-like receptor family pyrin domain containing 3 (NLRP3), suppressed infiltration of activated astrocytes, and reduced infarct volume [42]. On the motor function test, however, the effect of monotherapy with exosomes was uncertain, whereas combination therapy with exosomes and rosuvastatin significantly improved the proportion of left-side swings at seven days after stroke. This study concluded that BMSC-generated exosomes in combination with rosuvastatin promoted neuroprotection and suppressed cell death and neuroinflammation, thereby improving early functional recovery in rats [42].

### 5.2. Amplification of Specific Molecules in MSC-Generated Exosomes

Xin and colleagues used exosomes engineered with specific miRNA cluster genes. They studied whether exosomes derived from multipotent MSCs mediate miR-133b transfer that promotes neurological recovery after stroke in vivo, using knock-in and knock-down technologies to up-regulate and down-regulate miR-133b, respectively, by lentiviral infection in MSCs. In 2017, using this technology, they examined the effects of intra-arterial treatment of exosomes, and they extended the period for evaluating motor function to 28 days after tMCAO [43]. On the mNSS and the foot-fault test, miR-133b^+^ MSC-derived exosome treatment showed significant improvement of motor recovery from 7 to 28 days, and they overcame the effect of exosomes from BMSCs infected with blank vector from 14 to 28 days. In the peri-infarct area, the miR-133b-enriched exosomes further increased neurite remodeling and synaptogenesis in the peri-infarct area. In cultured astrocytes, as compared with naive exosome treatment, miR-133b-enriched exosomes significantly increased exosomes released by OGD astrocytes, whereas exosomes of miR-133b^-^ BMSCs significantly decreased such release. In addition, exosomes harvested from OGD astrocytes treated with exosomes of miR-133b^+^ BMSCs significantly increased neurite branching and elongation of cultured cortical embryonic rat neurons. Furthermore, Rab9 effector protein with kelch motifs (RABEPK), a molecule requiring endosomes for trans-Golgi network (TGN) transport, was decreased after treatment with exosomes of miR-133b^+^ BMSCs in vivo and in vitro. Thus, intra-arterial treatment with exosomes of miR-133b^+^ BMSCs released exosomes from astrocytes, possibly by downregulating RABEPK expression, and improved stroke recovery together with neurite regeneration [43]. In their next study, the mir-17-92 cluster, including miR-17, miR-18a, miR-19a, miR-20a, miR-19b, and miR-92a, was shown to enhance axonal outgrowth [20]. Overexpression of the miR-17-92 cluster for BMSC-derived exosomes significantly improved the mNSS and the foot-fault test. Administration of tailored BMSC exosomes containing elevated miR-17-92 cluster at 24 h after MCAO significantly improved motor function on the mNSS and the foot-fault test from 7 to 28 days. Neurogenesis and oligodendrogenesis, together with dendrite arborization and axonal outgrowth via the PTEN/Akt/mTOR and the GSK-3β pathways, were related to improvement of motor function in MCAO rats [44].

In the study by Liu et al., enkephalin and transferrin were found to be overexpressed in BMSC-generated exosomes, because enkephalin exerts neuroprotection, and transferrin receptor was highly expressed in the BBB [48,49]. BMSC-generated exosomes in which transferrin and enkephalin were overexpressed (enkephalin-Tar-exo) were assembled and administered for a transient MCAO rat model. On behavioral assessments, neurological scores were decreased, and the holding angle in the inclined board test was increased at one and three weeks after injection of enkephalin-Tar-exo in tMCAO compared to vehicle-treated rats. Levels of lactate dehydrogenase, p53, caspase-3, and nitric oxide in cerebrospinal fluid were decreased after treatment with enkephalin-Tar-exo. In the ischemic core and peri-infarct area, neuronal density was significantly increased in tMCAO rats with enkephalin-Tar-exo treatment compared with the vehicle group [45]. Thus, overexpression of enkephalin and transferrin receptors resulted in a suitable drug delivery system in the BBB, and it exerts neuroprotection against ischemic injuries.

Geng et al. examined the therapeutic effect of adipose-derived stem cells (ADSCs), and they focused on the promising effects of exosomal miR-126 that was poorly expressed in the plasma of stroke patients [46], and overexpression and knock-down of miR-126 for ADSCs were used. In rats subjected to tMCAO for 2 h, mNSS scores and motor function on the foot-fault test were significantly improved in the order of treatment with miR-126-overexpressed exosomes, naïve exosomes, exosomes with miR-126 knock-down, and vehicle treatment. Neurogenesis and angiogenesis were further enhanced by ADSC-derived miR-126-overexpressed exosomes, together with suppression of microglial activation and TNF-α and IL-1β production after ischemia [46].

## 6. Therapeutic Effect of Exosomes Compared with Cell Therapy

Several studies compared the therapeutic effect of exosomes with that of original cells. Exosomes derived from MSCs improved neurological function and enhanced neurogenesis and angiogenesis, comparable to the effect of human MSCs [37]. Another study showed that human MSC-derived exosomes showed similar neurological improvements to those of MSC-treated mice. Additionally, neuroprotection and neurogenesis were facilitated after treatment with human MSC-derived exosomes and human MSCs [37]. Chen and colleagues studied the therapeutic effects of ADMSCs from mini-pigs and of ADMSC-derived exosomes in rats subjected to tMCAO for 50 min. Of the sham-operated rats, the vehicle-treated rats, ADMSC-treated rats, ADMSC-derived exosome-treated rats, and both ADMSC- and ADMSC-derived exosome-treated rats, sensorimotor function on the corner test showed significant improvements in both ADMSC- and ADMSC-derived exosome-treated rats, ADMSC-treated rats, and ADMSC-derived exosome-treated rats compared to the vehicle-treated rats at 7, 14, and 28 days after tMCAO. Infarct areas were larger in the order of the vehicle-treated rats, ADMSC-derived exosome-treated rats, ADMSC-treated rats, and both ADMSC- and ADMSC-derived exosome-treated rats. Protein levels of inflammatory markers such as inducible nitric oxide synthase, TNF-α, and IL-1β, markers of oxidative stress, apoptosis, and fibrosis, as well as cellular expression of brain damage, inflammation, and brain edema, were suppressed by treatment with both ADMSCs and ADMSC-derived exosomes, ADMSCs, and ADMSC-derived exosomes compared to the vehicle groups [47]. Thus, infarct area was reduced, and neurological function was improved with treatment by xenogenic ADMSCs or ADMSC-derived exosomes. Furthermore, combination therapy with xenogenic ADMSCs and ADMSC-derived exosomes offered additional benefit over each monotherapy for reducing brain infarct volume and improving neurological function [47]. Thus, exosomes derived from both allogenic and xenogenic MSCs might have therapeutic efficacy, comparable to original MSCs.

## 7. Discussion

In the current study, a systematic review to explore the therapeutic effects of exosome administration for stroke recovery was conducted. The current review showed that most preclinical studies were based on rodent models of tMCAO, using MSCs or stem cells as sources of exosomes, with evaluations by various behavioral examinations up to 28 days after stroke. In particular, exosomes derived from MSCs or stem cells showed significant improvements of motor function from the subacute phase and chronic phase of stroke through neurogenesis, angiogenesis, axonal outgrowth and myelination, synaptogenesis, and suppression of microglia, together with anti-inflammatory effects (Figure 2). Furthermore, exosomes demonstrated equally therapeutic effects to therapies involving original cells, and amplification of specific molecules in the exosomes further enhanced treatment effects.

Stroke is the leading cause of disability, and approximately 40% of stroke survivors still showed poor functional outcomes after endovascular therapy and rt-PA [6]. So far, preclinical studies have shown the efficacy of MSCs for improving functional outcomes after stroke [15,16,17]. The theoretical strategy for cell therapies was based on replacement by differentiation of grafted MSCs or stem cells of dead neurons, but many studies showed that cell therapies promote regeneration and repair of damaged tissue due to stroke, including angiogenesis, neurogenesis, and axonal outgrowth, together with regulating corresponding molecular cascades [32]. Although the approach with these cell therapies was shown to be safe, and they passed through the BBB and reached injured tissues [50], one cannot exclude the possibility of complications after grafting cells, such as cell rejection, unpredictable immune responses, possible contamination, problems of storage of cells before their use, and inducing tumor formation [51,52,53,54]. Exosomes offer an advantage to overcome the limitations of cell therapy, such as: (1) less tumorigenicity due to their to inability to self-replicate; (2) dosing not affected by loss of transplanted cell viability; (3) not occluding the microvascular system; (4) tough lipid bilayer vesicles that can be easily stored for a long time at −80 °C and other extremes of handling, while retaining bioactivity; (5) their low immunogenicity not requiring a host immune response and a match between the donor and the recipient; (6) capable of extracting sufficient amounts from immortalized stem cells; and (7) capable of enhancing efficacy by genetic engineering of the parent cells [44,55,56,57]. Importantly, exosomes have an ability to cross the BBB [32]. MSC-derived exosomes were detected in infarct tissues after systemic administration, as well as in glial cells and neurons in vivo, indicating that MSC-derived exosomes successfully pass through the BBB and are taken up by endocytosis in cells in ischemic brains [41,58,59,60]. On the other hand, the current systematic review provided another crucial insight, showing that the therapeutic effects of MSC-derived exosomes for stroke recovery were comparable to the effects of therapy with original MSCs [37,47], suggesting that MSC-derived exosomes, rather than direct interaction of MSCs with brain repair, are responsible for restorative effects of MSCs after stroke. Collectively, exosome therapy could be considered as a potential candidate for a novel stroke therapy. However, the overwhelming majority of the evidence currently available deals with treatment effects of MSCs compared with exosomes [18,19], and thus further studies to explore the therapeutic efficacy of exosomes after stroke are warranted.

An increasing number of studies have explored exosomes other than MSCs or stem cells, which contributed to neuroregeneration and brain repair after stroke. In our previous study, inhibition of Semaphorin 3A in the subacute stage of stroke suppressed the activation of astrocytes and downregulated miR-30c-2-3p and miR-326-5p in astrocyte-derived exosomes, which were capable of enhancing axonal elongation in ischemic neurons by increasing prostaglandin D2 synthase [13]. Another study demonstrated that prion proteins in astrocyte-derived exosomes increased after ischemia, which exerted neuroprotection in vitro [61]. Xin et al. showed that therapeutically administered MSC-derived exosomes enriched with miR-133b stimulated the release of astrocyte-derived exosomes, which increased neurite outgrowth [43]. There is evidence of crosstalk between microglia and endothelial cells. Polarization of microglia induced by IL-4 increased miR-26a in microglia-generated exosomes, which may promote tube formation in vitro and angiogenesis in vivo after ischemia [62]. Thus, exosomes exerted direct and indirect effects for repairing and regenerating brains after stroke.

Exosomes deliver a specific composition of proteins, lipids, RNA, and DNA and they can work as cargo to transfer this information from donor cells to target cells, which presumably alters cell functions. In particular, microRNAs were enriched in exosomes, and they were shown to lead to reprogramming of the recipient cells and enhancement of neuroregeneration and brain repair after stroke. Argonaute2 (Ago2) protein, a component of the RNA-induced silencing complex that binds miRNAs and facilitates mRNA degradation, is required for exosome-promoted axonal outgrowth, and deletion of Ago2 in MSC-derived exosomes suppressed axonal outgrowth [63]. Studies included in our systematic review demonstrated that exosomes from USCs enhanced neurogenesis related to transfer of miR-26a for HDAC6 inhibition, and miR-184 and miR-210 in MSC-generated exosomes were associated with neurogenesis and angiogenesis [38,45]. Furthermore, tailored exosomes with amplification of the miR-17-92 cluster facilitated neurogenesis, oligodendrogenesis, and axonal outgrowth via the PTEN/Akt/mTOR and GSK-3β pathways, which were previously shown to be key mechanisms in stroke recovery [13,14,21], and those with miR-133b increased neurite remodeling and synaptogenesis by regulating RABEPK, according to the studies of Xin et al. [20,44]. Overexpression of miR-126 in ADSC-derived exosomes further enhanced not only neurogenesis and angiogenesis, but also suppression of microglial activation and TNF-α and IL-1β production after stroke [46]. Recent evidence suggests that immunological processes were implicated in the mechanisms of brain repair after stroke [64,65]. Thus, amplification of specific cargo in exosomes can pleiotropically enhance therapeutic effects leading to brain repair and neuroregeneration.

On the other hand, overexpression of enkephalin and transferrin in MSC-generated exosomes showed robust neuroprotection after ischemia, because enkephalin has a neuroprotective effect, and transferrin receptors were highly expressed in the BBB [45]. In another in vitro study, the cyclo (Arg-Gly-Asp-D-Tyr-Lys) peptide [c(RGDyK)] conjugated on the surface of MSC-derived exosomes was engineered. Because c(RGDyK) has high affinity to integrin α _v_ β _3_ in reactive cerebral vascular endothelial cells after ischemia [66,67], c(RGDyK)-conjugated exosomes (cRGD-Exo) targeted the lesion region of the ischemic brain after intravenous administration in the tMCAO. Although cRGD-Exo also accumulated in the liver and lung, an increase in targeting of cRGD-Exo for infarcted brain compared to unmodified exosomes was greater than the corresponding increase in the liver and lungs. Importantly, cRGD-Exo efficiently entered microglia, neurons, and astrocytes and reduced the protein levels of TNF-a, IL-1b, and IL-6, as well as suppressed microglial activation and neural apoptosis [58]. Thus, it is promising to amplify specific cargo leading to regulation of signaling pathways in recipient cells, as well as to develop drug delivery systems to attach in the endothelium and pass through the BBB, which can promote the therapeutic effects of exosomes for stroke recovery.

The potential limitations of the current study must be considered when interpreting this systematic literature review. In the current study, preclinical studies focusing on stroke recovery that were evaluated by behavior analyses and related molecular mechanisms were examined, and studies without such physiological analyses were excluded. Thus, there could be a potential bias, and the systematic literature review did not cover the overall mechanisms of exosomes for stroke recovery.

In conclusion, exosomes play a pivotal role in neuroregeneration and brain repair after stroke, and tailored exosomes with specific molecules could further enhance therapeutic efficacy. So far, only a few clinical trials have been conducted in stroke patients, and the therapeutic efficacy and safety of exosomes are essentially unknown in humans. However, the current review showed that exosomes can be potential therapeutic candidates for improving functional recovery in stroke survivors. We hope that our study helps to provide insight and a future perspective on the potential use of exosomes.

## Figures and Tables

**Figure 1 ijms-21-06894-f001:**
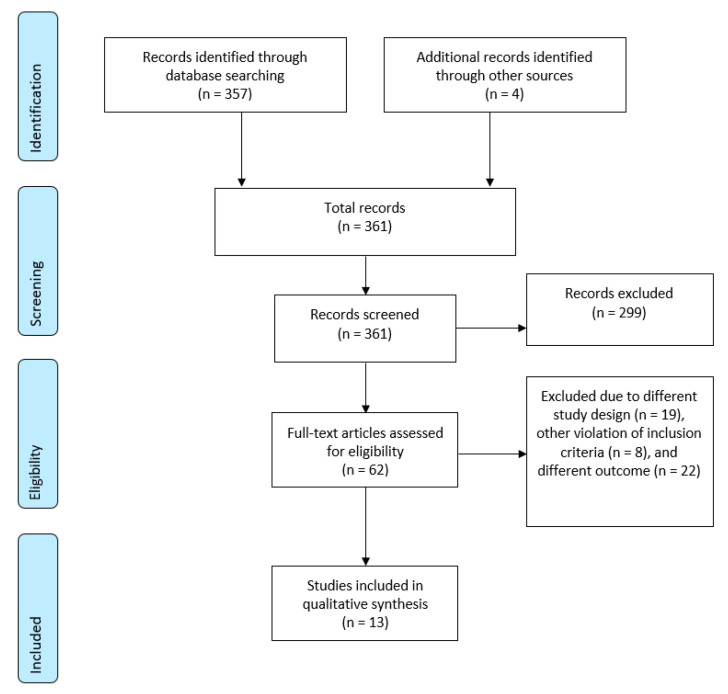
Flow diagram of the systematic literature search. Preferred Reporting Items for Systematic Reviews and Meta-Analyses (PRISMA) flow diagram shows the number of records identified, included, and excluded through the different phases of a systematic review.

**Figure 2 ijms-21-06894-f002:**
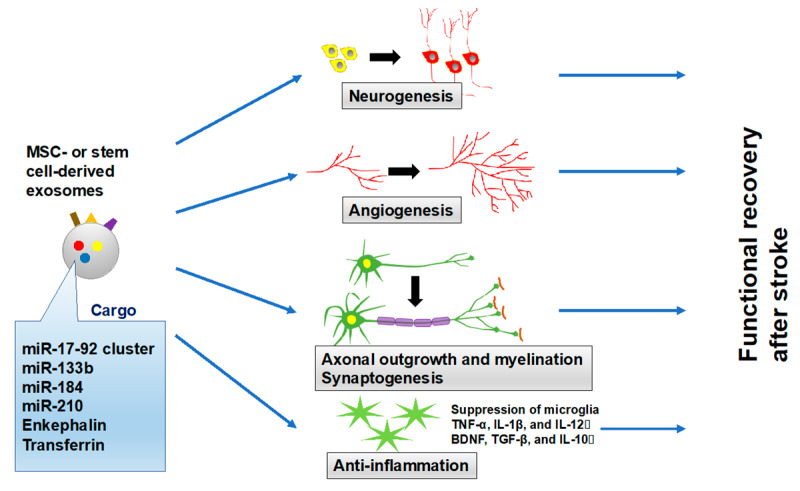
Effects of MSC-derived or stem cell-derived exosomes on stroke recovery. Exosomes derived from MSCs or stem cells show significant improvements of motor recovery through neurogenesis, angiogenesis, axonal outgrowth and myelination, synaptogenesis, and suppression of microglia, together with anti-inflammatory effects. MSC = Mesenchymal stromal cell.

**Table 1 ijms-21-06894-t001:** Characteristics of preclinical studies of exosome treatment for stroke.

Authors, Year	Species in Experiments In Vivo	Stroke Model, Duration of Ischemia, Min	Therapeutic Intervention by Exosomes, Route of Administration, Timing, Dosage	Source of Exosome	Behavioral Outcome Assessment	Maximum Date of Evaluation for Motor Function
Xin H et al., 2013 [35]	Male Wistar rats (weighing 270–300 g)	tMCAO, 120	IV injection, 24 h after ischemia, 100 μg	BMSCs	1. mNSS 2. Foot-fault test	28
Zhao Y et al., 2020 [36]	Male SD rats (weighing 270 ± 10 g)	tMCAO, 90	IV injection, 2 h after ischemia, 120 μg	BMSCs	1. Neurological severity score 2. Shuttle-box test	7
Doeppner TR et al., 2015 [37]	Male C57BL/6 mice (10 weeks old)	tMCAO, 30	IV injection, 3 and 5 days after ischemia, 2 x10^6^ MSCs released	BMSCs	1. Rotarod test 2. Tightrope test 3. Corner turn test	28
Ling X et al., 2020 [38]	Male SD rats (6–8 weeks old, weighing 250–300 g)	tMCAO, 120	IV injection, 4 h after ischemia, approximately 1 × 10^11^	Urine-derived stem cells	1. mNSS 2. Foot-fault test	28
Nalamolu KR et al., 2019 [39]	Male SD rats (weighing 210 ± 10 g)	tMCAO, 120	IV injection, immediately after reperfusion, 150 μg *	HUCB-MSCs	1. mNSS 2. Modified adhensive removal test 3. Beam walking 4. Accelerating rotarod performance tests	7
Nalamolu KR et al., 2019 [40]	Male SD rats (weighing 210 ± 10 g)	tMCAO, 120	IV injection, immediately after reperfusion, 150 μg **	HUCB-MSCs	1. mNSS 2. Modified adhensive removal test 3. Beam walking 4. Accelerating rotarod performance tests	7
Moon Gj et al., 2019 [41]	Male SD rats (8 weeks old, 270–300 g)	tMCAO, 90	IV injection, 24 h after ischemia, 30 μg	BMSCs	1. mNSS 2. Cylinder and ladder rung walking test	28
Safakheil M et al., 2020 [42]	Male Wistar rats (weighing 280–300 g)	tMCAO, 60	Stereotaxic administration, 3 h after ischemia; 100 μg, oral gavage, rosuvastatin (20 mg/kg/day); or both	BMSCs	1. The elevated body swing test 2. Garcia score	7
Xin H et al., 2017 [43]	Male Wistar rats (weighing 270–300 g)	tMCAO, 120	IV injection, 24 h after ischemia, 100 μg	BMSCs	1. mNSS 2. Foot-fault test	28
Xin H et al., 2017 [44]	Male Wistar rats (weighing 270–300 g)	tMCAO, 120	IV injection, 24 h after ischemia, 100 μg (comparable to 3 X 10^11^ particles)	BMSCs	1. mNSS 2. Foot-fault test	28
Liu Y et al., 2019 [45]	SD rats (8–12 weeks old, weighting 220–240 g)	tMCAO, 120	IV injection, 4 or 12 h after ischemia, 0.5 × 10^5^ particles	BMSC	1. Neurological scores 2. Inclined board test	21
Geng W et al., 2019 [46]	Male SD rats (weighing 280 ± 10 g)	tMCAO, 120	IV injection, 24 h after ischemia, exosome pellet in 200 μL,	Rat adipose derived stem cells	1. mNSS 2. Foot-fault test	14
Chen KH, et al., 2016 [47]	Male SD rats (weighing 350–375 g)	tMCAO, 50	IV injection, 3 h after ischemia, 100 μg	ADMSC	1. Corner test	28

SD = Sprague Dawley, tMCAO = transient middle cerebral artery occlusion, IV = intravenous, MSCs = mesenchymal stromal cells, BMCSs = bone marrow mesenchymal stromal cells, ADMSC = adipose derived mesenchymal stem cell, mNSS = modified neurological severity score. * = exosomes derived from normal MSCs, ** = combination of exosomes derived from normal and hypoxic MSCs.

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
