# Peer review of "Pleiotropic Effects of Exosomes as a Therapy for Stroke Recovery"

_ijms, 2020, doi:10.3390/ijms21186894_

Round 1
Reviewer 1 Report
In this systematic literature review, the authors discuss the effects of exosomes on ischemic stroke outcome and highlight the role of specific microRNAs and molecular mechanisms that mediate and enhance therapeutic effects such as neurogenesis, angiogenesis, axonal outgrowth, and synaptogenesis in the ischemic brain. Detailed methods for study selection including search criteria (key words), PRISMA guidelines, exclusion criteria, are clearly provided and 13 studies have been included in the review. The findings of these 13 studies have been reported in detail and the effects of cell therapy compared to exosome therapy. These studies provide strong evidence favoring exosome therapy for the treatment of stroke. This review is very well written with a complementary use of tables and figures. Overall, this is an excellent review and I only have few minor comments:
- Please expand all abbreviations at first mention.
- Section 5.2, Geng et al study: miR126+ exosomes, the ‘+’ sign is not legible.
- In the section comparing cell therapy and exosome therapy, the authors have summarized the findings of the study and both ADMSCs and ADMSC-exosomes decrease infarct area and improve outcome compared to vehicle group. It is important to know and discuss if ADMSC-exosomes induced significantly better therapeutic effects compared to ADMSCs?
Author Response
In this systematic literature review, the authors discuss the effects of exosomes on ischemic stroke outcome and highlight the role of specific microRNAs and molecular mechanisms that mediate and enhance therapeutic effects such as neurogenesis, angiogenesis, axonal outgrowth, and synaptogenesis in the ischemic brain. Detailed methods for study selection including search criteria (key words), PRISMA guidelines, exclusion criteria, are clearly provided and 13 studies have been included in the review. The findings of these 13 studies have been reported in detail and the effects of cell therapy compared to exosome therapy. These studies provide strong evidence favoring exosome therapy for the treatment of stroke. This review is very well written with a complementary use of tables and figures. Overall, this is an excellent review and I only have few minor comments:
- Please expand all abbreviations at first mention.
Reply: We appreciate this comment. We have expanded the following abbreviations: TNF-α, IL-1β, BDNF, TGF-β, ERK, NLRP1, NLRP3, mTOR, LDH, NO, CSF, iNOS, and RISC.
Line 24, page 7: BMSC-derived exosomes also significantly suppressed M1 microglial polarization and increased M2 microglial cells, together with reduction of pro-inflammatory cytokines, such as tumor necrosis factor (TNF)-α, interleukin (IL)-1β, and IL-12, and upregulation of anti-inflammatory molecules, such as brain-derived neurotrophic factor, transforming growth factor-β, and IL-1, and thereby downregulated extracellular signal regulated kinase 1/2 phosphorylation [36].
Line 29, page 9: Monotherapy with exosomes, as well as combination therapy, decreased cell death together with downregulation of NOD-Like Receptor Family Pyrin Domain Containing 1 (NLRP1) and NOD-Like Receptor Family Pyrin Domain Containing 3(NLRP3), suppressed infiltration of activated astrocytes, and reduced infarct volume [42].
Line 15, page 11: Levels of lactate dehydrogenase, p53, caspase-3, and nitric oxide in cerebrospinal fluid were decreased after treatment with enkephalin-Tar-exo.
Line 17, page 12: Protein levels of inflammatory markers such as inducible nitric oxide synthase, TNF-α, and IL-1b, markers of oxidative stress, apoptosis, and fibrosis, as well as cellular expression of brain damage, inflammation, and brain edema, were suppressed by treatment with both ADMSCs and ADMSC-derived exosomes, ADMSCs, and ADMSC-derived exosomes compared to the vehicle groups [49].
Line 32, page14: Argonaute2 (Ago2) protein, a component of the RNA-induced silencing complex that binds miRNAs and facilitates mRNA degradation, is required for exosome-promoted axonal outgrowth, and deletion of Ago2 in MSC-derived exosomes suppressed axonal outgrowth [63].
- Section 5.2, Geng et al study: miR126+ exosomes, the ‘+’ sign is not legible.
Reply: Thank you for the comment. We revised our manuscript as follows.
Line 25, page 11: In rats subjected to tMCAO for 2 h, mNSS scores and motor function on the foot-fault test were significantly improved in the order of treatment with miR-126-overexpressed exosomes, naïve exosomes, exosomes with miR-126 knock-down, and vehicle treatment. Neurogenesis and angiogenesis were further enhanced by ADSC-derived miR-126-overexpressed exosomes, together with suppression of microglial activation and TNF-α and IL-1β production after ischemia [48].
- In the section comparing cell therapy and exosome therapy, the authors have summarized the findings of the study and both ADMSCs and ADMSC-exosomes decrease infarct area and improve outcome compared to vehicle group. It is important to know and discuss if ADMSC-exosomes induced significantly better therapeutic effects compared to ADMSCs?
Reply: The study by Chen et al (ref no. 49) showed that therapeutic interventions were more effective in the order of both ADMSCs and ADMSC-exosomes, ADMSCs, and ADMSC-exosomes for improving motor function, reducing infarct area, and altering any biomarkers. In this study, ADMSC-exosomes were not more effective than original ADMSCs.
Reviewer 2 Report
This systematic review is a useful tool when assessing all available evidence of exosome use for stroke. Whilst this treatment has a long way to go until we can fully assess it's usefulness in the clinic, this review demonstrates the mounting and significant evidence which supports its use and need for more research in the field.
Please find my comments and suggestions below to the authors:
Introduction
- There is a lack of context provided on where the research is up to for exosome use in stroke (in-vitro, preclinical, clinical). Need to emphasise in abstract and introduction that you were only looking at preclinical research for the systematic review. You were guided by the available evidence and therefore the lack of mention of clinical research was due to the lack of evidence at a clinical trial level. To help with comprehensiveness, i recommend including in your introduction the link between where research is up to clinically versus preclinically for exosomes.
Methods - In your methods there are no protocols/methods mentioned for the data extracted and then how this data was handled. This makes up a significant amount of your results and needs to be added. You will need to re-work and add a greater level of detail to your methods.
- There is no reference to table 1 or Fig 2 in text. This needs to be rectified in your results.
Discussion
Overall the discussion needs to be revised to maintain correctness with literature on use of exosomes as well as the benefits over whole cell-based therapies.
- The theoretical strategy for cell therapy use in stroke also now includes immune-modulation to support brain repair. This will need to be included in your discussion as some of the points are based around immunogenicity and ability to engraft as potential limitations of cell therapies over exosomes
- MSCs have been shown to cross the blood brain barrier- particularly in models of stroke/HI where the blood brain barrier may be comprimised. Some points in your discussion will need to be contextualised to maintain correctness
- Please note that exosomes are not essential for cell-cell interactions in ischemic brains. Astrocyte-induced release of exosomes can be stimulated by therapeutically administered MSC-derived exosomes
Other comments:
- There is a high-level of self-citation in this paper (at least 7 of your own papers). I would ensure that when you refer to these research findings, these are placed in more context. i.e. reference to study 13 helps to allude to the mechanisms of axonal outgrowth after application of exosomes. But this is one example of mechanisms involved with exosome application and no other mechanisms are explored in this much detail.
- There is a typo in the discussion- mid page 9. Should read in-vivo not in-vitro.
Author Response
This systematic review is a useful tool when assessing all available evidence of exosome use for stroke. Whilst this treatment has a long way to go until we can fully assess it's usefulness in the clinic, this review demonstrates the mounting and significant evidence which supports its use and need for more research in the field.
Please find my comments and suggestions below to the authors:
Introduction
There is a lack of context provided on where the research is up to for exosome use in stroke (in-vitro, preclinical, clinical). Need to emphasise in abstract and introduction that you were only looking at preclinical research for the systematic review. You were guided by the available evidence and therefore the lack of mention of clinical research was due to the lack of evidence at a clinical trial level. To help with comprehensiveness, i recommend including in your introduction the link between where research is up to clinically versus preclinically for exosomes.
Reply: We thank the reviewer for this important comment. We focused on functional recovery after stroke in animal stroke models, because exosome therapy has not been applied in a clinical setting. We added the following sentences.
Line 5, page 5: So far, exosome therapy has not been applied in a clinical setting, only a few clinical trials have been conducted in stroke patients (https://clinicaltrials.gov); thus, the therapeutic efficacy and safety of exosomes are essentially unknown in humans. Therefore, a systematic review to explore the treatment effects of exosomes for functional recovery after stroke in preclinical studies using animal stroke models was performed.
Methods
In your methods there are no protocols/methods mentioned for the data extracted and then how this data was handled. This makes up a significant amount of your results and needs to be added. You will need to re-work and add a greater level of detail to your methods.
Reply: Thank you for the comment. In the current study, we conducted a systematic review according to the PRISMA guideline. As demonstrated in Figure 1, ‘Identification’, ‘Screening’, and ‘Eligibility’ were done appropriately, and 13 studies were finally included. However, as suggested by the reviewer, the main and additional outcomes were not stated in the Methods section. Thus, we have added them.
Line 21, page 5: The main outcome measures that have been considered were mainly related to motor function and neurological deficits. In part, additional outcomes, such as infarct area, molecular pathways including microRNA, mRNA, and protein, and inflammatory markers or systemic cytokine levels, have also been assessed. Because the current systematic literature search focused on functional stroke recovery after exosome therapy, in vitro and in vivo studies without a physiological analysis were excluded.
There is no reference to table 1 or Fig 2 in text. This needs to be rectified in your results.
Reply: I am sorry for this error. We have now cited Table 1 (line 17, page 6) in the text, with citation of Fig. 2 on line 9, page 13. Thank you.
Discussion
Overall the discussion needs to be revised to maintain correctness with literature on use of exosomes as well as the benefits over whole cell-based therapies.
Reply: We are grateful for the reviewer’s comment, with which we agree. Some studies examined whether exosomes had equal therapeutic effects for stroke recovery to original cells. However, evidence for exosomes was insufficient compared to that for MSCs, and further studies are needed.
Line 3, page 14: On the other hand, the current systematic review provided another crucial insight, showing that the therapeutic effects of MSC-derived exosomes for stroke recovery were comparable to the effects of therapy with original MSCs [37, 49], suggesting that MSC-derived exosomes, rather than direct interaction of MSCs with brain repair, are responsible for restorative effects of MSCs after stroke. Collectively, exosome therapy could be considered as a potential candidate for a novel stroke therapy. However, the overwhelming majority of the evidence currently available deals with treatment effects of MSCs compared with exosomes [18, 19], and thus further studies to explore the therapeutic efficacy of exosomes after stroke are warranted.
The theoretical strategy for cell therapy use in stroke also now includes immune-modulation to support brain repair. This will need to be included in your discussion as some of the points are based around immunogenicity and ability to engraft as potential limitations of cell therapies over exosomes
Reply: Thank you for this important comment. Recent evidence suggests that immunological processes were associated with brain repair after stroke. Exosome therapy, by regulating immunological alterations after stroke, can play an important role in stroke recovery. We have revised our manuscript as follows, and referred to the following 2 papers.
Line 12, page 15: Overexpression of miR-126 in ADSC-derived exosomes further enhanced not only neurogenesis and angiogenesis, but also suppression of microglial activation and TNF-α and IL-1β production after stroke [48]. Recent evidence suggests that immunological processes were implicated in the mechanisms of brain repair after stroke [64, 65]. Thus, amplification of specific cargo in exosomes can pleiotropically enhance therapeutic effects leading to brain repair and neuroregeneration.
Reference
- Iadecola, C.; Anrather, J., The immunology of stroke: from mechanisms to translation. Nat Med 2011, 17, (7), 796-808.
- Jian, Z.; Liu, R.; Zhu, X.; Smerin, D.; Zhong, Y.; Gu, L.; Fang, W.; Xiong, X., The Involvement and Therapy Target of Immune Cells After Ischemic Stroke. Front Immunol 2019, 10, 2167.
MSCs have been shown to cross the blood brain barrier- particularly in models of stroke/HI where the blood brain barrier may be comprimised. Some points in your discussion will need to be contextualised to maintain correctness
Reply: We appreciate this excellent comment. As suggested, it has been shown that MSCs cross the BBB; we therefore revised our manuscript as follows, and we added a reference (no.50).
Line 19, page 13: Although the approach with these cell therapies was shown to be safe, and they passed through the BBB and reached injured tissues [50], one cannot exclude the possibility of complications after grafting cells, such as cell rejection, unpredictable immune responses, possible contamination, problems of storage of cells before their use, and inducing tumor formation [51-54].
Reference no. 50: Conaty, P.; Sherman, L. S.; Naaldijk, Y.; Ulrich, H.; Stolzing, A.; Rameshwar, P., Methods of Mesenchymal Stem Cell Homing to the Blood-Brain Barrier. Methods Mol Biol 2018, 1842, 81-91.
Please note that exosomes are not essential for cell-cell interactions in ischemic brains.
The release of astrocyte-derived exosomes can be stimulated by therapeutically administered MSC-derived exosomes enriched with miR-133b.
Reply: As suggested by the reviewer, we revised our manuscript as follows.
Line 20, page 14: Xin et al showed that therapeutically administered MSC-derived exosomes enriched with miR-133b stimulated the release of astrocyte-derived exosomes, which increased neurite outgrowth [43].
Line 26, page 14: Thus, exosomes exerted direct and indirect effects for repairing and regenerating brains after stroke.
Other comments:
There is a high-level of self-citation in this paper (at least 7 of your own papers). I would ensure that when you refer to these research findings, these are placed in more context. i.e. reference to study 13 helps to allude to the mechanisms of axonal outgrowth after application of exosomes. But this is one example of mechanisms involved with exosome application and no other mechanisms are explored in this much detail.
Reply: Thank you for this comment. As noted, there is a high-level of self-citation in this paper. We have reduced it to 4 papers (ref nos. 13,14,20,21) studying the mechanisms of axonal outgrowth after ischemia, and we moved these references to the Introduction section. Additionally, in the Discussion section, we noted that PTEN/Akt/mTOR and GSK-3β pathways were shown as key mechanisms of axonal outgrowth in the above papers, and they were also implicated in the mechanisms of axonal outgrowth after exosome therapy. We revised our manuscript as follows.
Line 8, page 4: Previous studies examined whether Phosphatase and Tensin Homolog Deleted from Chromosome (PTEN)/Akt/Glycogen Synthase Kinase 3beta (GSK-3β) signaling was implicated in the mechanisms of axonal outgrowth in rodent ischemic brains [14, 20]. Furthermore, we demonstrated that inhibition of semaphoring 3A from the acute to the subacute phase of stroke in the peri-infarct area increased axonal outgrowth related to the Rho family GTPase 1 (Rnd1)/R-Ras/Akt/GSK-3β pathway, together with suppression of glial fibrillary acidic protein (GFAP)-immunoreactive astrocytes, and functional recovery was promoted in the chronic stage of stroke [13]. Additionally, it was shown that PTEN/Akt/mammalian target of rapamycin(mTOR) signaling was implicated in the mechanisms of axonal outgrowth after L-carnitine therapy in a rat model of chronic cerebral hypoperfusion [21].
Line 7, page 15: Furthermore, tailored exosomes with amplification of the miR-17-92 cluster facilitated neurogenesis, oligodendrogenesis, and axonal outgrowth via the PTEN/Akt/mTOR and GSK-3β pathways, which were previously shown to be key mechanisms in stroke recovery[13, 14, 21], and those with miR-133b increased neurite remodeling and synaptogenesis by regulating RABEPK, according to the studies of Xin et al [20, 44].
There is a typo in the discussion- mid page 9. Should read in-vivo not in-vitro.
Reply: Thank you. We revised our manuscript as follows.
Line 32, page 13: MSC-derived exosomes were detected in infarct tissues after systemic administration, as well as in glial cells and neurons in vivo, indicating that MSC-derived exosomes successfully pass through the BBB and are taken up by endocytosis in cells in ischemic brains [41, 58-60].